# Unsupervised Clustering of Neighborhood Associations and Image Segmentation Applications

**Zhenggang Wang [1,2,3,*], Xuantong Li [3], Jin Jin [1,2], Zhong Liu [2] and Wei Liu [3]**

[1] Chengdu Computer Application Institute Chinese Academy of Sciences, Chengdu 610041, China; jinjin@nsu.edu.cn

[2] University of Chinese Academy of Sciences, Beijing 100864, China; gsdxl2015@163.com

[3] Chengdu Customs of China, Chengdu 610041, China; thustoms@163.com (X.L.); liuwei@customs.gov.cn (W.L.)

[*] Correspondence: wangzhenggang@customs.gov.cn

**Abstract:** Irregular shape clustering is always a difficult problem in clustering analysis. In this paper, by analyzing the advantages and disadvantages of existing clustering analysis algorithms, a new neighborhood density correlation clustering (NDCC) algorithm for quickly discovering arbitrary shaped clusters. Because the density of the center region of any cluster sample dataset is greater than that of the edge region, the data points can be divided into core, edge, and noise data points, and then the density correlation of the core data points in their neighborhood can be used to form a cluster. Further more, by constructing an objective function and optimizing the parameters automatically, a locally optimal result that is close to the globally optimal solution can be obtained. This algorithm avoids the clustering errors caused by iso-density points between clusters. We compare this algorithm with other five clustering algorithms and verify it on two common remote sensing image datasets. The results show that it can cluster the same ground objects in remote sensing images into one class and distinguish different ground objects. NDCC has strong robustness to irregular scattering dataset and can solve the clustering problem of remote sensing image.

**Keywords:** irregular shape cluster; automatic parameter optimization; neighborhood connection; point density; remote sensing clustering

## 1. Introduction

Cluster analysis is the most commonly used static data analysis method. Cluster analysis refers to the process of grouping a collection of physical or abstract objects into multiple classes composed of similar objects. The objects in the same cluster have great similarity, while objects in different clusters have great divergence. In general, clustering methods can be divided into mean-shift, density-based, hierarchical, spectral clustering [1], and grid-based [2] methods.

Different algorithms have different advantages and problems. Centroid-based algorithms, such as K-means (Kmeans) [3,4], K-medoid [5,6], fuzzy c-means (FCM), Mean shift [7,8], and some improved methods [9,10], have the advantages of simple principles, convenient implementation, and fast convergence. Because this kind of algorithm always takes the approach of finding the centroid and clustering the points close to the centroid, they are especially suitable for clustering. Such algorithms have the characteristics of good clustering results and low time complexity. However, real clustering samples usually contain a large number of clusters of arbitrary shapes. Consequently, centroid-based clustering

algorithms, which cluster the points around a centroid into one class, lead to poor results on irregular shape clusters and many misclassified points.

Clusters of arbitrary shapes can be easily detected by a method that is based on local data point density. The density-based spatial clustering of applications with noise (DBSCAN) [11] has good robustness for clusters with uniform density of any shape. However, it is not easy to select a suitable threshold. Especially for clusters with large differences in density, the threshold selection is very difficult. Moreover, the circular radius needs to be adjusted constantly in order to adapt to different cluster densities, and there is no reference. At the same time, for clusters without obvious boundaries, it is easy to classify two clusters with different classes as belonging to the same class. Because DBSCAN uses the global density threshold MinPts, it can only find clusters that are composed of points with a density that satisfies this threshold; that is, it is difficult to find clusters with different densities. Moreover, clustering algorithms that are based on hierarchy, spectral features, and density also have serious difficulties with parameter selection.

In real clustering problems, there are many clusters with arbitrary shapes, and it is impossible to use a center of mass in order to represent the nature of the data in the cluster. Moreover, not all of the data have a real clustering center; and, in some cases, the centroid points of clusters with completely different distributions basically coincide, so clustering data based on centroid points often leads to misjudgments. Parameter selection that is based on the density-based algorithms is also very difficult, which often causes poor clustering results. No matter what kind of clustering algorithm, there are difficulties in parameter selection. Besides, methods, such as Silhouette Coefficient and sum of the squared errors, cannot completely realize unsupervised parameter selection. Our aims are to avoid the shortcomings of centroid- and density-based algorithms, and address the challenges of clustering datasets with clusters having different point densities.

The neighborhood density correlation clustering (NDCC) does not use the method of calculating the centroid within the cluster. Instead, it incorporates the idea of density clustering, but is not limited to the density of a fixed region and does not use a certain definite distance or a certain definite density as a measure of the differentiation between different classes. It takes the $k$ nearest neighbor domain of each point as the analysis object, and considers each point and its neighboring $k$ points as the same cluster data. By adjusting the k value, different clustering results are obtained. Although a series of parameters can be manually set, it is difficult to find suitable parameters for clustering without sufficient prior knowledge and multiple trials. By appropriate objective function setting and minimizing the objective function, the NDCC can automatically adjust the parameters to get the optimal solution and automatically cluster a sample dataset. The method can detect irregular shape clusters and automatically find the correct number of clusters. This method does not consider the influence of iso-density points between clusters in cluster classification. Its generalization performance and robustness are improved (iso-density points between clusters in this paper are similar to the noise points or edge points connecting two clusters in other work. This is because the existence of iso-density points between clusters will lead to misclassification when using a density-based clustering algorithm to distinguish these clusters).

**The contributions of this paper are as follows:**

- By relying on the adaptive distance radius to distinguish core, edge, and noise points, the method that is proposed in this paper overcomes the problem that a distance threshold is difficult to select in a density-based clustering algorithm, eliminates the influence of some non-core points on the clustering process, and enhances the generalization performance of the algorithm.
- Neighborhood density correlation is used instead of a real distance in order to measure the correlation between the core points. In addition, the method clusters a certain number of neighboring core points around a core point as a class. This approach can adapt to clustering problems with different density clusters in the same dataset.

- An appropriate objective function is adopted in order to minimize the distance between some core points, achieve the local optimal clustering results, completely avoid the subjective factors of manually set parameters, improve the efficiency and objectivity of the algorithm, and realize unsupervised clustering.

## 2. Related Work

In addition to the above centralization algorithm and density-based algorithm, in order to evaluate the effect of the algorithm proposed in this paper on data point clustering, we consulted a large number of literatures on clustering algorithm.

The Gaussian mixture model (GMM) clustering [12,13] algorithm is equivalent to a generalization of Kmeans and other algorithms, and it can form clusters of different sizes and shapes. The characteristics of the data can be better described with only a few parameters. However, the amount of computation that is needed for the GMM algorithm is large and it converges slowly. Therefore, Kmeans is usually used to preprocess the sample set, and the initial values of the GMM is determined according to the obtained clusters.

Hierarchical clustering (HC) methods, such as Balanced Iterative Reducing and Clustering using Hierarchies (BIRCH) [14], Robust Clustering using linKs (ROCK) [15], and Chameleon [16], compute the distances between samples first. They next merge the closest points into the same class each time. Subsequently, the distance between the classes is calculated, and the nearest class is merged into a larger class. The merging continues until a class is synthesized. HC has the advantages that the similarity of the distance and rules are easy to define, the hierarchy of classes can be determined, and it can deal with any shape of clusters. However, because of its high computational complexity, it is easily affected by singular values, and different numbers of clustering levels will lead to different clustering results. However, the purpose of clustering is to divide the data into certain categories. The number of clustering layers of HC is difficult to select, and it is difficult to evaluate the clustering results with an objective function.

In recent years, several new clustering algorithms have been widely used in the field of data analysis. Fop et al. [17] proposed a new clustering algorithm for mixed models and introduced a mixed version of the Gaussian covariance graph model for sparse covariance matrix clustering. In this method, a likelihood penalty is adopted for estimation. A penalty term on the graph structure is used in order to induce different degrees of sparsity, and prior knowledge is introduced. A structural electromagnetic algorithm is used for parameter estimation and graph structure estimation, and two alternative strategies that were based on a genetic algorithm and efficient stepwise search were proposed.

OPTICS [18] ranks neighborhood points to identify the clustering structure in order of density, and then visualizes clusters of different densities. OPTICS must be able to find clusters by looking for "valleys" in a visualization graph created by other algorithms, so its performance is directly constrained by these algorithms. Moreover, it cannot complete the clustering process completely unsupervised.

Moraes et al. [19] proposed a data clustering method that was based on principal curves. The *k*-segment algorithm uses the extracted principal curves in order to complete the data clustering process.

Abin et al. [20] investigated learning problems for constrained clustering and proposed a supervised learning-based method to deal with different problems in constrained clustering. Linear and non-linear models were considered, improving the clustering accuracy.

Rodriguez et al. [21] proposed a method that clusters according to whether the center of the cluster density is higher than its neighbors. In their method, high density and a relatively large distance between points are used in order to complete the clustering process. Although it is simple, their approach can better solve the problem of arbitrary shaped clusters. However, as with DBSCAN, choosing the distance threshold parameter value is difficult. A value that is too large or too small will affect the clustering results.

Distributed clustering based on density and hesitant fuzzy clustering methods have a certain progress. Corizzo et al. [22] put forward a kind of DENCAST system for sensor networks, in order to be able to solve the single objective regression and multiple regression task goal. It performs density clustering on multiple computers, but it does not require a final merge step, which breaks through the traditional mode of distributed clustering. Hosseini et al. [23] proposed a new dense-based soft clustering method that was based on the Apache Spark computing model, which is mainly used for the new hesitant fuzzy weighted similarity measurement of gene expression, especially suitable for the clustering problem of large data sets. In recent years, the clustering problem based on deep learning network has attracted people's attention. Zhao et al. [24] proposed introducing a multi-task learning framework based on CNN, which combines self-supervised learning and task of scene classification. The classification accuracy of NWPU, AID and other four data sets reached more than 90%. Petrovska et al. [25] used a deep architecture of two streams, while using support vector machines (SVM) to classify tandem features. The experimental results show that this method has certain competitive advantages.

NDCC is proposed due to the existence of manual setting of hyper-parameters in existing clustering algorithms, and it being difficult to give consideration to spherical cluster and irregular shape cluster clustering at the same time. In this paper, seven scatter-point data sets are verified, and various indicators obtained good results. Furthermore, the 'UCMerced-LandUse' remote sensing dataset and '2015 high-resolution remote sensing image of a city in southern China dataset' are compared with other algorithms, and the clustering accuracy rate reached approximately 90%. The algorithm flow is shown in Figure 1. The Summary of notations is showed in the Appendix A Table A1.

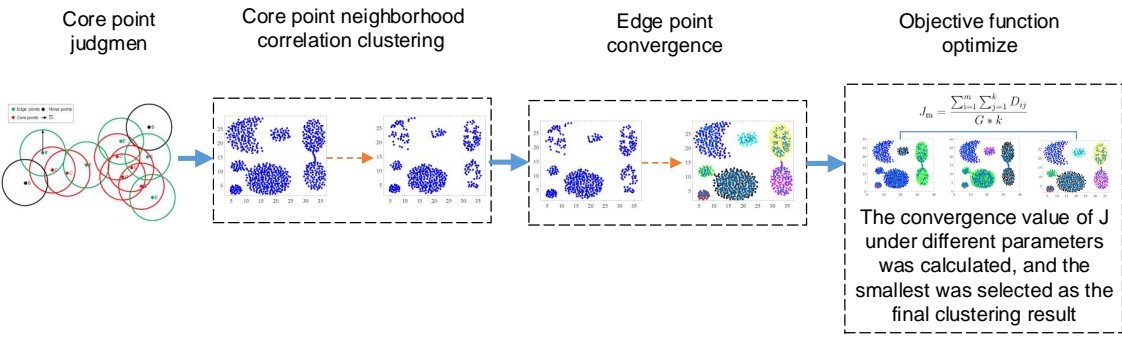

**Figure 1.** The process of the neighborhood density correlation clustering (NDCC) algorithm.

## 2.1. Core Point Judgment

If there are clusters in a given dataset, then the data points of the same kind of cluster can generally be divided into core data points (referred to as core points) and edge data points (referred to as edge points). An edge point is a point on the edge of the cluster, and a core point is a data point inside the cluster. The density of data points in any direction around a core point is relatively high, whereas the density of data points around the edge point is relatively high only in the direction toward the core point. In general, the number of points around the core point is more than three as many as that around the edge point in the data cluster. With this feature, the number of points within a certain radius determines whether the point is an edge point or a core point, so that the edge points and core points of a dataset can be easily divided. Figure 2 shows the schematic diagram of core points, edge points, and noise points.

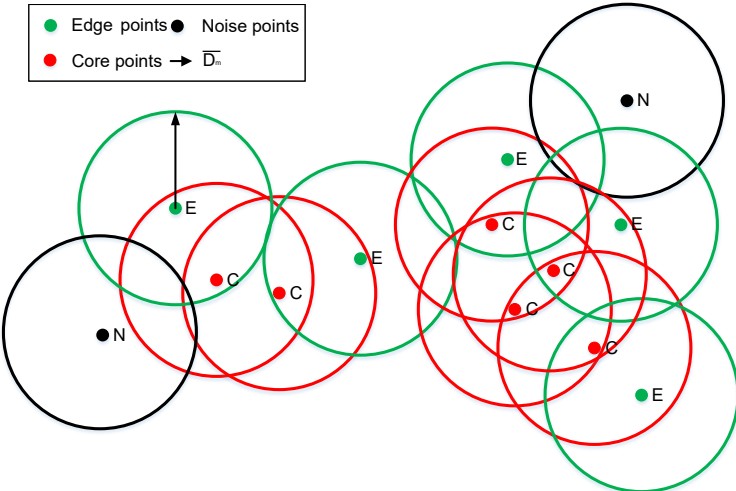

**Figure 2.** Distribution of data points with *m* equal to three. Within a distance radius of *m* in the ordered distance matrix *S*, the number of neighborhood points of the red point is greater than *m*, so it is a core point; the number of neighborhood points of the green point is greater than zero and less than *m*, so it is an edge point; and, the number of neighborhood points of the black point is equal to 0, so it is a noise point.

The distance from each data point to other data points in the sample set is calculated in order to constitute the distance matrix from all data point to points, and the distance matrix is sorted in ascending order. We can obtain an ordered distance matrix *S* and ordered serial number matrix *A* (which stores the serial number of the points of *S*). A quick sort method (e.g., heap or merge sort) is adopted for sorting. In order to determine the nature of a data point, first, a circle centered on the point with a radius that is equal to the mean of the distance to the *m*th point in the overall ordered distance matrix *S* is found (the mean is denoted as $\overline{D_m}$). If there is only one point inside the circle, then this point is a noise point, and if the number of points inside the circle is more than one and less than or equal to *m*, then it is an edge point (note that the number of points in the circle and the radius parameter of the circle in *S* are both *m*, and these values should be consistent). Otherwise, if there are more than *m* points, then the point is a core point (in contrast to DBSCAN and other density clustering algorithms, the radius here is set adaptively to avoid the radius threshold selection difficulty). The cluster of core points can better reflect the shape of the original cluster. After removing the edge points and noise points, the number of dataset points is greatly reduced. Moreover, the factors that interfere with clustering are removed, which is more conducive to clustering the dataset (the core points' ordered serial number matrix $A'$ takes the form of *A* and the number of neighborhood points of the *i*-th point $x_i$ in the dataset is defined as $N_i(x_i)$). The determination formula of the core point is as follows:

$$N_i(x_i) = \{x_i \in D \mid D(i,j) \leqslant \overline{D_m}\} \tag{1}$$

$$x_i \in \begin{cases} E_i, N_i(x_i) \leqslant m \\ C_i, N_i(x_i) > m \\ B_i, N_i(x_i) = 0 \end{cases} \tag{2}$$

Here, $E_i$ is an edge point, $C_i$ is a core point, $D(i,j)$ is the distance between $x_i$ and $x_j$, and $B_i$ is a noise point.

### 2.2. Core Point Neighborhood Correlation Clustering

We can adopt two different strategies, compact and sparse, in order to deal with the clustering of core points within the cluster. Different strategies will cluster sample sets into different numbers of clusters, and different clustering results can be obtained by adjusting the clustering strategy. When a compact strategy is adopted, a strong connection between two points is needed in order to classify them into one class. Hence, the sample set is clustered into a large number of independent clusters. When a sparse strategy is adopted, only a weak connection between two points is needed, and the sample set is clustered into a small number of independent clusters. NDCC does not consider the impact of noise points for the time being.

When neighborhood correlation clustering is adopted, as long as the appropriate strategies are adopted, core point clusters of arbitrary density can be found and a better clustering result can be achieved. The steps of the neighborhood correlation clustering method used in this paper are as follows. First, take the pre-$k$-dimension data points in the core point ordered serial number matrix $A'$ as the analysis object. The pre-$k$ neighbor points that are closest to the core data point $C_i$ are grouped into the same cluster. The calculation that is used to group points is similar to the process of bacterial infection. An infection needs a medium and, for each core $C_i$, the core of the $k$-nearest neighbor in $A'$ is the medium. Other points can be absorbed into the cluster through the medium. Subsequently, the algorithm iterates through all of the core data points until it is not possible to merge a new core data point to form a cluster. The process of the scattered-point data aggregation class is shown in Figure 3. Among them, Figure 3a is the original distribution of the scatter diagram, Figure 3b is the distribution of core points, and Figure 3c is the clustering result of the scatter diagram.

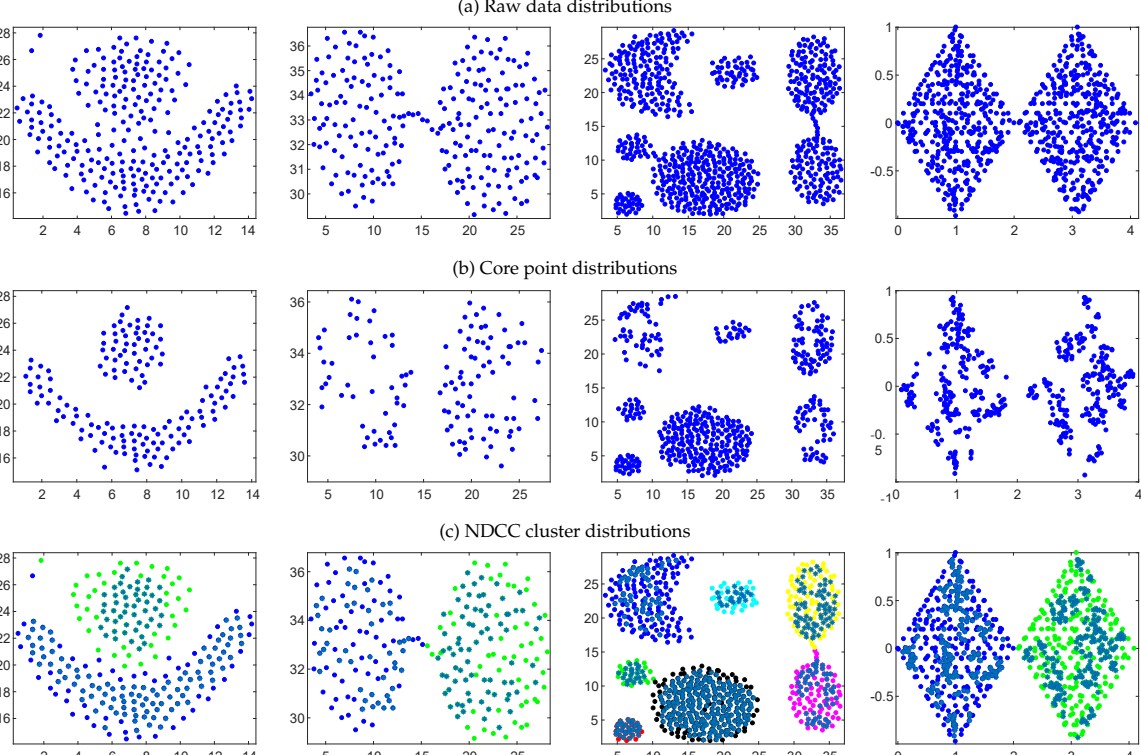

**Figure 3.** Sample datasets: (**a**) raw data, (**b**) core point, and (**c**) NDCC cluster distributions: From left to right the Flame, Brige, Aggregation, and Two Diamonds datasets are shown. The core points are marked with ∗. Because the dataset has no noise points, the situation in which no noise points are set is shown.

### 2.3. Edge Point Convergence

After the core point clustering is completed, the set of core point clusters can be obtained. All of the edge points are grouped into the nearest cluster of core points according to their distance from the core. Unlike other clustering algorithms [26] that optimize the objective function iteratively, the allocation of cluster edge points is performed in a single step, and the allocation of edge points does not affect the clustering distribution of the core points.

### 2.4. Objective Function

Although the data points of a cluster group have the same property to a certain extent, in fact, sometimes points with relatively large distances will be grouped into one group. It has been proved that parameter optimization is an effective method to solve the optimal clustering in many unsupervised learning [27]. If we use the features of all points in the cluster to measure the clustering result, there will be a large deviation that can be avoided by using a local average density evaluation model. The clustering effect of sample data can be measured by the density compactness of local points, and the local compactness of each point can be estimated by the sum of the distance from the data point to the adjacent points. If the ratio of the sum of distances from each point to the $k$ nearest-neighbor points and the number of clustering $G$ and $N_i(x_i)$ are small, the density compactness of local points is high. In this paper, this index is defined as the local density compactness coefficient (LDCC). The minimum value of LDCC leads to the best clustering results. This coefficient is used as objective function $J$ in order to optimize parameters $m$ and $k$.

$$J_m = \frac{\sum_{i=1}^{m} \sum_{j=1}^{k} D_{ij}}{G * k} \tag{3}$$

Here, $G$ is the number of clusters and $D_{ij}$ is the distance from each point to $k$ neighboring core points.

The optimal classification cluster is determined by the minimum local average density. The value of $m$, $k$ increases from 2, so that the whole clustering process goes from fine to coarse. When all of the data points are grouped into a cluster, it suggests that, for the current values of $m$ and $k$, there is no effect in continuing to reinforce neighborhood correlation. When the dataset is clustered to one category, the upper limit of $m$ is equal to $N$ and the upper limit of $k$ is equal to $N1$. Thus, the combinations of $N$ and $N1$ values are uniquely determined by different datasets. At this time, the value of $k$ is fixed and $m$ is continue increased to obtain different values of $J$. When the data points are all grouped to one cluster, the stopping increase the $m$ value. Subsequently, increasing the $k$ value by step size 1 and repeating the above process. The minimum value of $J$ corresponds to the final values of $m$ and $k$. The result can be considered as a locally optimal clustering result that is close to the global optimum (if there are sufficient hardware resources, it is generally possible to obtain all the solutions of the objective function $J$ without setting the upper limit of parameters until the objective function reaches the globally optimal solution. However, if the values of $m$ and $k$ are too large, all data points will be clustered into one cluster, which has no practical significance). On the seven data sets, the corresponding $J$ value in the process of $m$ and $k$ value growth is shown in Table 1. In the table for this article, Compound is abbreviated as Com and Aggregation is abbreviated as Agg.

**Table 1.** Different values of $m$ and $k$ corresponding to the value of objective function $J$.

|  | m\k | 1 | 2 | 3 | 4 | 5 | 6 | 7 | 8 | 9 |
|---|---|---|---|---|---|---|---|---|---|---|
| Flame | 1 | NaN | 95.486 | NaN | NaN | NaN | NaN | NaN | NaN | NaN |
|  | 2 | NaN | 44.232 | 10.290 | 3.1983 | 1.8581 | 2.0600 | 2.2541 | 2.4422 | 2.6205 |
|  | 3 | NaN | 44.040 | 17.200 | 1.5365 | 1.7507 | 1.9392 | 2.1237 | 2.3023 | 2.4707 |
|  | 4 | NaN | 31.116 | **1.2558** | 1.5135 | 1.7232 | 1.9112 | 2.0920 | 2.2683 | 2.4343 |
|  | 5 | NaN | 41.397 | 1.3853 | 1.4392 | 1.6448 | 1.8254 | 1.9999 | 2.1695 | NaN |
|  | 6 | NaN | 32.542 | 1.3901 | 1.4433 | 1.6473 | 1.8281 | 1.9993 | 2.1673 | 2.3259 |
| Com | 1 | NaN | NaN | NaN | NaN | NaN | NaN | NaN | NaN | NaN |
|  | 2 | NaN | 126.08 | 56.315 | 20.601 | 23.522 | 26.391 | 29.027 | 32.075 | 34.803 |
|  | 3 | NaN | 178.02 | 57.112 | 21.465 | 24.470 | 27.482 | 30.237 | 32.844 | 35.776 |
|  | 4 | NaN | 121.79 | 55.200 | 20.030 | 22.934 | 26.258 | 29.189 | 31.913 | 34.350 |
|  | 5 | NaN | 117.35 | 55.413 | 19.947 | 22.835 | 25.626 | 28.700 | 31.533 | 34.076 |
|  | 6 | NaN | 129.69 | 60.642 | **19.252** | 21.953 | 25.164 | 28.018 | 30.657 | 33.045 |
| Agg | 1 | NaN | 387.29 | 62.412 | 16.160 | 11.760 | 13.206 | 14.525 | 15.732 | 16.879 |
|  | 2 | NaN | 184.33 | 233.26 | 73.421 | 30.284 | 28.637 | 17.937 | 19.475 | 20.916 |
|  | 3 | NaN | 253.21 | 168.26 | 11.359 | 13.265 | 14.920 | 16.373 | 17.711 | 18.969 |
|  | 4 | NaN | 182.86 | 200.88 | 62.268 | 24.675 | 27.767 | 23.998 | 26.048 | 28.257 |
|  | 5 | NaN | 347.17 | 165.87 | 49.315 | 20.053 | 22.498 | 24.706 | 26.772 | 28.714 |
|  | 6 | NaN | 178.39 | 106.21 | **10.078** | 18.811 | 21.103 | 23.166 | 25.097 | 26.912 |
| Jain | 1 | NaN | 187.03 | 67.867 | 25.657 | **1.6562** | 1.9069 | NaN | NaN | NaN |
|  | 2 | NaN | 167.72 | 48.871 | 43.765 | 4.6532 | 9.3520 | NaN | NaN | NaN |
|  | 3 | NaN | 144.48 | 54.808 | 16.929 | 7.2830 | 7.5172 | NaN | NaN | NaN |
|  | 4 | NaN | 135.61 | 41.909 | 39.149 | 4.8330 | NaN | NaN | NaN | NaN |
|  | 5 | NaN | 80.011 | 30.257 | 13.277 | 3.6233 | 2.9865 | 3.3977 | 3.7759 | NaN |
|  | 6 | NaN | 214.42 | 53.325 | 2.1711 | 2.8313 | 3.3713 | NaN | NaN | NaN |
| R15 | 1 | NaN | 177.49 | 46.660 | 4.5060 | 5.947 | 6.7796 | 7.5406 | 10.863 | 11.691 |
|  | 2 | NaN | 103.91 | 38.383 | 17.972 | 9.7477 | 6.2142 | 5.3470 | 5.8632 | 6.3426 |
|  | 3 | NaN | 66.870 | 33.183 | 16.920 | 5.2751 | 6.0342 | 6.743 | 7.4357 | 7.7695 |
|  | 4 | NaN | 82.606 | 25.579 | 12.964 | 4.9973 | 4.7220 | 5.8390 | 6.8188 | 7.8437 |
|  | 5 | NaN | 60.787 | 24.970 | 8.4691 | 4.9771 | 5.8233 | 6.5233 | 7.1739 | 7.9340 |
|  | 6 | NaN | 67.342 | 26.202 | 6.4120 | **4.5052** | 5.2823 | 6.1329 | 7.3036 | 8.3255 |
| D31 | 1 | NaN | NaN | NaN | NaN | NaN | NaN | NaN | NaN | NaN |
|  | 2 | NaN | 1866.6 | 681.63 | 501.13 | 177.65 | 102.19 | 70.809 | 58.494 | 38.216 |
|  | 3 | NaN | 969.08 | 863.35 | 381.16 | 139.83 | 71.810 | 53.632 | 58.509 | 42.448 |
|  | 4 | NaN | 1307.3 | 558.92 | 183.68 | 92.075 | 68.552 | 55.875 | 50.926 | 36.792 |
|  | 5 | NaN | 1347.0 | 575.20 | 169.27 | 87.900 | 38.450 | 43.441 | 50.049 | 39.394 |
|  | 6 | NaN | 1518.4 | 573.43 | 108.78 | 38.310 | **30.481** | 36.513 | 39.710 | 42.689 |
| Spiral | 1 | NaN | 124.03 | **1.2121** | 1.7720 | 1.4179 | NaN | NaN | NaN | NaN |
|  | 2 | NaN | 85.887 | 2.0650 | 2.4893 | 3.1583 | 2.1054 | NaN | NaN | NaN |
|  | 3 | NaN | 47.485 | 12.410 | 1.6699 | 2.0612 | 2.5540 | NaN | NaN | NaN |
|  | 4 | NaN | 241.71 | 1.3481 | 1.9870 | 2.4541 | 3.1556 | 3.7281 | 3.3966 | NaN |
|  | 5 | NaN | 87.836 | 1.3304 | 2.0597 | 2.5755 | 3.1968 | 2.4343 | 2.8320 | NaN |
|  | 6 | NaN | 84.506 | 1.4340 | 2.1149 | 2.6030 | 3.2090 | 3.7147 | 4.2960 | NaN |

The bold values in Table 1 are the minimum values of the objective function $J$, and the corresponding parameters $m$ and $k$ are the parameters of NDCC optimal clustering.

## 3. Experiment

The experiment in this paper consists of two parts. The first part is to compare the algorithms that are mentioned in this paper in the common clustering data set and show the visual effect and index difference. The second part compares the visual effect and index difference of the algorithm on two public remote sensing data.

### 3.1. Index

This paper compares the following indicators of different algorithms: Accuracy ($Acc$), Normalized Mutual Information ($NMI$), Rand Index ($RI$), Adjusted Rand Index ($ARI$), Mirkin index ($MI$), and Hubert Index ($HI$). The following symbols are independent, and they are not associated with the symbol in the paper above.

#### 3.1.1. *Acc*

The formula for calculating the *Acc* of sub-datasets is as follows:

$$Acc = \frac{\sum_{i=1}^{n} \delta\left(s_i, map\left(r_i\right)\right)}{n}$$

where $r_i$ and $s_i$ represent the obtained label and the real label corresponding to data point $x_i$, respectively; $n$ represents the total number of data points and $\delta$ represents the indicator function, as follows:

$$\delta(x, y) = \begin{cases} 1 & \text{if } x = y \\ 0 & \text{otherwise} \end{cases}$$

The *map* in the equation represents the optimal class object re-allocation, in order to ensure correct statistics.

#### 3.1.2. *NMI*

Mutual information is a useful measure in information theory. It can be regarded as the information that is contained in a random variable regarding another random variable, or the uncertainty reduced by a random variable due to the knowledge of another random variable. The formula of $NMI$ can be derived, as follows: Suppose tthat he $(X, Y)$ are two random variables with the same number of elements. The joint distribution of $(X, Y)$ is $P(x, y)$ and their marginal distributions are $P(x)$ and $P(y)$. Furthermore, $MI(x, y)$ is the mutual information and it is the relative entropy of joint distribution $P(x, y)$ and product distribution $P(x)(y)$. Therefore, we have

$$MI(X, Y) = \sum_{i=1}^{N} \sum_{j=1}^{N} P(x, y) \log\left(\frac{P(x, y)}{P(x)P(y)}\right) \tag{4}$$

Here, $P(x)$ is the probability distribution function of $X$ and $P(y)$ is the probability distribution function of $Y$. The joint probability distribution $P(x, y) = \frac{|x_i \cap y_j|}{N}$ while using the abovementioned formula can be expressed, as follows:

$$NMI(X, Y) = \frac{2MI(X, Y)}{H(x) + H(y)} \tag{5}$$

The distribution of $H(X)$ and $H(Y)$ is the entropy of information for the random variable $X$ and $Y$.

$$H(X) = -\sum_{i=1}^{N} P(x_i) \log(P(x_i)); H(Y) = -\sum_{j=1}^{N} P(y_j) \log\left(P(y_j)\right) \tag{6}$$

### 3.1.3. *RI*, *ARI*, *MI*, *HI* and *JC*

Let the clustering result be $C = \{C_1, C_2, \cdots, C_m\}$, and the known partition be $P = \{P_1, P_2, \cdots, P_m\}$, Rand Index (*RI*) [28], and Jacarrd Index (*JI*) [28]. Subsequently, we have the following:

$$RI = \frac{a+d}{a+b+c+d} \tag{7}$$

$$JI = \frac{a}{a+b+c} \tag{8}$$

where, $a$ indicates that the two data objects belong to the same cluster in $C$ and the same group in $P$; $b$ indicates that the two points belong to a cluster in $C$, but to different groups in $P$. $c$ indicates that the two points do not belong to the same cluster in $C$, while $P$ belongs to the same group of $d$, which in turn indicates that the two points do not belong to the same cluster in $C$ and are in different groups in $P$. The higher the evaluation value of these two indexes, the closer the clustering result is to the real partition result, and the better the clustering effect.

For the *ARI*, it is assumed that the distribution of the model is random, which is, the division of $P$ and $C$ is random. Consequently, the number of data points of each category and cluster is fixed.

$$ARI = \frac{RI - E(RI)}{\max(RI) - E(RI)} \tag{9}$$

$E(RI)$ refers to the mean value of each cluster $RI$ and $\max(RI)$ to the maximum value of each cluster $RI$.

*Acc* is a simple and transparent evaluation measure and *NMI* can be information-theoretically interpreted. The *RI* and *ARI* penalize both false positive and false negative decisions during clustering. The formulas for *MI* and *HI* are available in Lawrence Hubert's paper [28]. While the larger of these index, including *Acc*, *NMI*, *RI*, *ARI*, *JI*, and *HI* represent the better clustering. Smaller *MI* represents better clustering, and *MI* is used as a reverse index to evaluate the performance of the algorithm.

### 3.2. Effect Evaluation of Scattered-Point Data Clustering

Six different algorithms are used to complete the clustering experiment on seven two-dimensional public datasets include Flame [29], Jain [30], Spiral [31], Aggregation [32], Compound [33], D31 [31], and R15 [31]. NDCC adopted the completely unsupervised objective function LDCC convergence method that was proposed by us to complete the clustering, and the other algorithms used manual parameter tuning in order to achieve better clustering effect as far as possible. From the experimental results, NDCC can complete clustering in a better way without intervention, and the effect is better than other algorithms. The indexes comparison are shown in Table 2 and the display effect of clustering is shown in Figure 4.

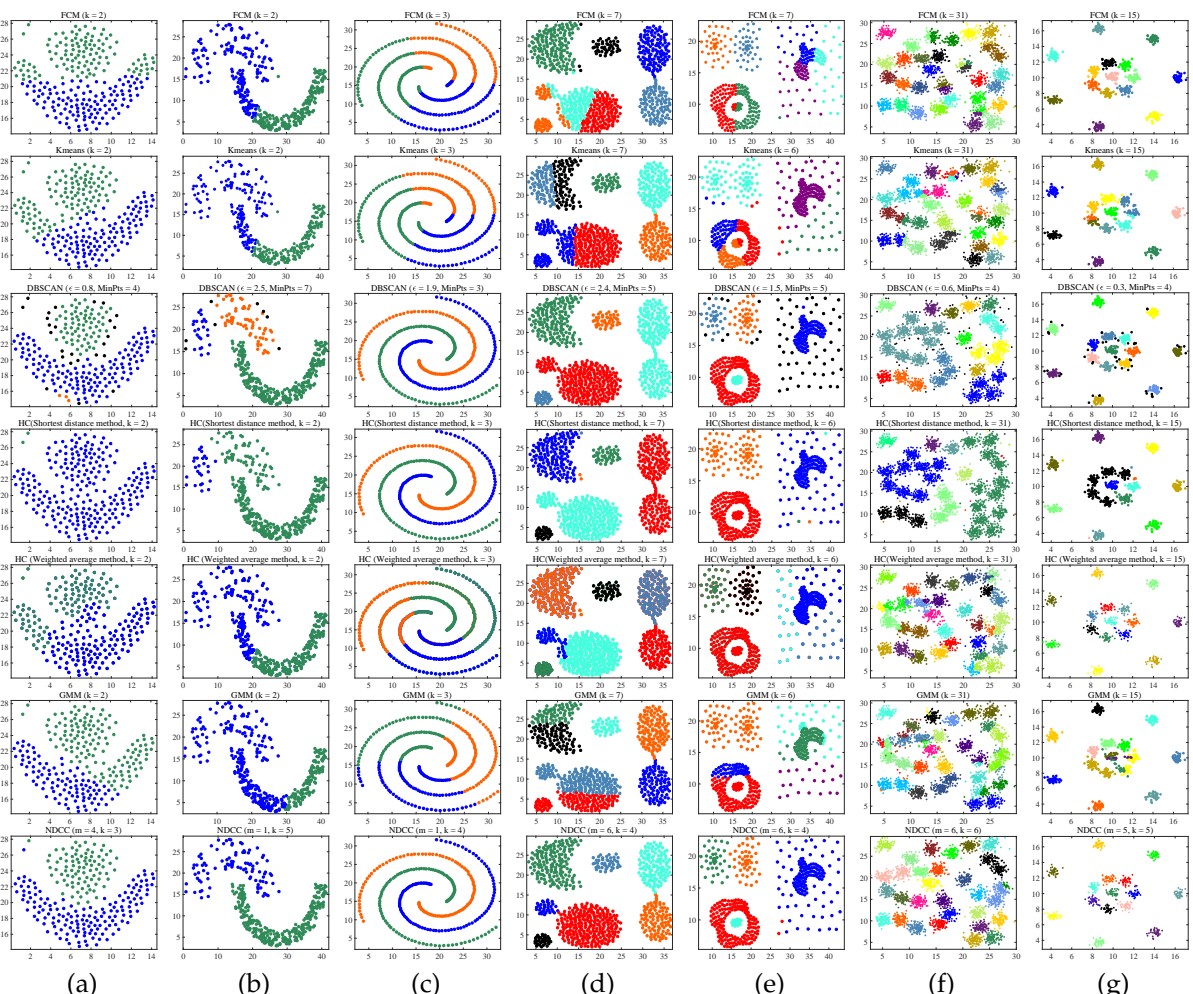

**Figure 4.** Clustering effects of various algorithms on seven different datasets. Shown from top to bottom are results for FCM, Kmeans, DBSCAN, HC1: shortest-distance HC, HC2: weighted average HC, GMM, and NDCC on (**a**) Flame, (**b**) Jain, (**c**) Spiral, (**d**) Aggregation, (**e**) Compound, (**f**) D31, and (**g**) R15.

**Table 2.** Indexes and Time (s) of various algorithms on seven datasets. HC1: shortest-distance HC, HC2: weighted HC.

| Algorithms | Index | Flame | Com | Agg | Jain | R15 | D31 | Spiral |
|---|---|---|---|---|---|---|---|---|
| NDCC | *NMI* | 0.82351 | 0.92465 | 0.98415 | 1.00000 | 0.99165 | 0.92821 | 1.00000 |
| | *ARI* | 0.91766 | 0.91539 | 0.99198 | 1.00000 | 0.98981 | 0.84395 | 1.00000 |
| | *RI* | 0.95903 | 0.92554 | 0.99728 | 1.00000 | 0.99868 | 0.98946 | 1.00000 |
| | *MI* | **0.04097** | **0.07445** | **0.00271** | **0.00000** | **0.00131** | **0.01054** | **0.00000** |
| | *HI* | 0.91806 | 0.85108 | 0.99457 | 1.00000 | 0.99737 | 0.97892 | 1.00000 |
| | Time | 0.13370 | 0.19083 | 0.53052 | 0.19714 | 0.16911 | 0.89410 | 0.12913 |
| Kmeans | *NMI* | 0.39153 | 0.76214 | 0.79032 | 0.38651 | 0.83125 | 0.85685 | −0.00352 |
| | *ARI* | 0.43119 | 0.52155 | 0.62437 | 0.30035 | 0.64740 | 0.70675 | 0.00566 |
| | *RI* | 0.71552 | 0.83216 | 0.88502 | 0.65009 | 0.94638 | 0.97907 | 0.55429 |
| | *MI* | 0.28448 | 0.16784 | 0.11498 | 0.34991 | 0.05362 | 0.02093 | 0.44571 |
| | *HI* | 0.43103 | 0.66431 | 0.77005 | 0.30018 | 0.89277 | 0.95814 | 0.10858 |
| | Time | 0.36484 | 0.42385 | 0.47220 | 0.45956 | 0.44716 | 0.91085 | 0.56192 |

**Table 2.** *Cont.*

| Algorithms | Index | Flame | Com | Agg | Jain | R15 | D31 | Spiral |
|---|---|---|---|---|---|---|---|---|
| DBSCAN | *NMI* | 0.76254 | 0.92354 | 0.89532 | 0.87512 | 0.90245 | 0.78925 | <u>1.00000</u> |
|  | *ARI* | 0.80492 | <u>0.96348</u> | 0.80894 | 0.93322 | 0.85614 | 0.33659 | <u>1.00000</u> |
|  | *RI* | 0.90171 | <u>0.98636</u> | 0.92730 | 0.96786 | 0.98292 | 0.89823 | 1.00000 |
|  | *MI* | 0.09829 | 0.01364 | 0.07270 | 0.03214 | 0.01708 | 0.10177 | 0.00000 |
|  | *HI* | 0.80342 | <u>0.97272</u> | 0.85460 | 0.93571 | 0.96584 | 0.79647 | 1.00000 |
|  | Time | 0.07515 | 0.03241 | 0.03049 | 0.07944 | 0.12112 | 0.46979 | 0.12087 |
| FCM | *NMI* | 0.42915 | 0.72424 | 0.78132 | 0.38125 | 0.99123 | 0.90125 | −0.01231 |
|  | *ARI* | 0.48796 | 0.53569 | 0.61123 | 0.30035 | 0.97277 | 0.78198 | 0.00625 |
|  | *RI* | 0.74393 | 0.84275 | 0.88286 | 0.65009 | 0.99912 | 0.98562 | 0.55415 |
|  | *MI* | 0.25607 | 0.15725 | 0.11714 | 0.34991 | 0.00188 | 0.01438 | 0.44585 |
|  | *HI* | 0.48787 | 0.68550 | 0.76572 | 0.30018 | 0.99824 | 0.97123 | 0.10829 |
|  | Time | 0.13276 | 0.14221 | 0.26508 | 0.15811 | 0.21768 | 0.63627 | 0.16774 |
| GMM | *NMI* | 0.41242 | 0.83451 | 0.78534 | −0.20412 | 0.93453 | 0.87967 | 0.02245 |
|  | *ARI* | 0.31724 | 0.70309 | 0.56320 | 0.00983 | 0.90361 | 0.66096 | 0.00707 |
|  | *RI* | 0.65914 | 0.90932 | 0.85984 | 0.51036 | 0.98767 | 0.97505 | 0.52445 |
|  | *MI* | 0.34087 | 0.09068 | 0.14016 | 0.48964 | 0.01233 | 0.02495 | 0.47555 |
|  | *HI* | 0.31827 | 0.81864 | 0.71968 | 0.02073 | 0.97535 | 0.95010 | 0.04889 |
|  | Time | 0.13361 | 0.31808 | 0.34967 | 0.10267 | 0.38937 | 0.59863 | 0.12543 |
| HC1 | *NMI* | 0.08453 | 0.80645 | 0.90127 | 0.28145 | 0.84352 | 0.68356 | <u>1.00000</u> |
|  | *ARI* | 0.01275 | 0.74248 | 0.80421 | 0.25629 | 0.54246 | 0.17390 | <u>1.00000</u> |
|  | *RI* | 0.54062 | 0.89035 | 0.92568 | 0.69094 | 0.90972 | 0.77892 | 1.00000 |
|  | *MI* | 0.45938 | 0.10965 | 0.07432 | 0.30906 | 0.09028 | 0.22108 | 0.00000 |
|  | *HI* | 0.08124 | 0.78071 | 0.85136 | 0.38188 | 0.81943 | 0.55785 | 1.00000 |
|  | Time | 0.07683 | 0.08276 | 0.12791 | 0.41091 | 0.10257 | 0.46250 | 0.02506 |
| HC2 | *NMI* | 0.39541 | 0.71354 | 0.78521 | 0.41568 | 0.98515 | 0.88145 | 0.03125 |
|  | *ARI* | 0.35635 | 0.84791 | 0.95488 | 0.36154 | 0.98565 | 0.75724 | 0.00269 |
|  | *RI* | 0.67866 | 0.94049 | 0.98498 | 0.68108 | 0.99825 | 0.98397 | 0.55328 |
|  | *MI* | 0.32134 | 0.08951 | 0.01502 | 0.31892 | 0.00175 | 0.01604 | 0.44672 |
|  | *HI* | 0.35732 | 0.88098 | 0.96997 | 0.36216 | 0.99651 | 0.96793 | 0.10656 |
|  | Time | 0.09683 | 0.19276 | 0.32451 | 0.81091 | 0.31348 | 0.66857 | 0.13546 |

The underlined values are the maximum *NMI*, *ARI*, *RI* and *HI* value. The bold values is the minimum value of *MI*.

## 3.3. Evaluation of Clustering Effect of Remote Sensing Data

In the field of remote sensing, it is expensive and difficult to obtain labeled data for training. Different ground features and different weather conditions make remote sensing images substantially different. Thus, it is difficult to apply a supervised learning method. In contrast, an unsupervised machine learning algorithm does not need training samples. It can cluster the data according to their natural distribution characteristics that are based on the spectral information given by geomagnetic radiation intensity in remote sensing images. It is a great way to group similar objects together. In this paper, we select several effective unsupervised clustering algorithms when compared with NDCC on two datasets. These are the labeled remote sensing dataset 'UCMerced-LandUse' [34], and the '2015 high-resolution remote sensing image of a city in southern China' [35] dataset. The evaluation of the two datasets is divided into two steps: preprocessing and evaluation.

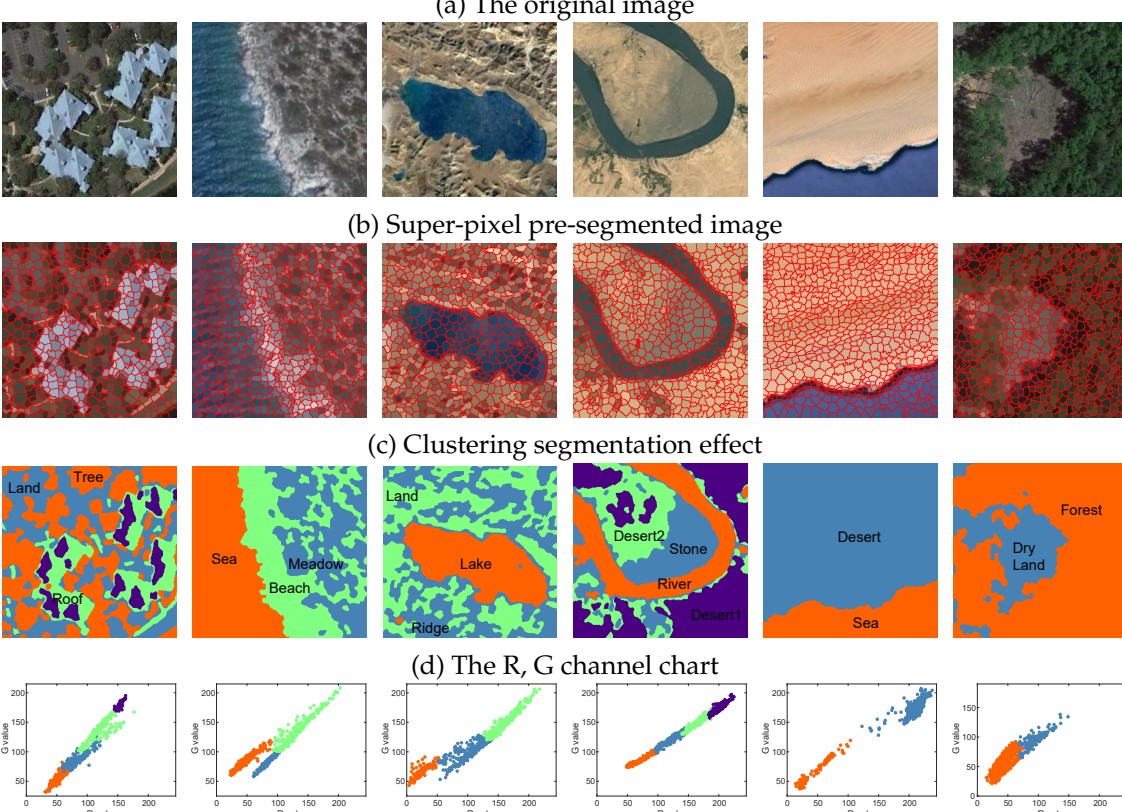

**Figure 5.** NDCC algorithm on the 'UCMerced-LandUse' remote sensing dataset clustering segmentation effect display. (**a**) are the original image. (**d**) are the distribution of superpixel scatter points of different types of ground objects in R and G channel graphs, corresponding to the segmentation of different ground objects in (**c**) graph. Our algorithm finds the number of image clusters in a completely unsupervised manner and realizes clustering segmentation.

**Step 1 Preprocessing**: super-pixel segmentation (the simple linear iterative clustering super-pixel segmentation algorithm [36–38]) is adopted as a pre-processing step for remote sensing image clustering to reduce the amount of calculation. The number of super-pixel elements in each images is kept between 1000 and 3000. Figure 5c,d show the image and scatter effect of NDCC remote sensing clustering. It can be seen from the Figure 5d that the distribution of the super-pixel data points in remote sensing images presents an irregular shape, no definite clustering center.

**Step 2 Evaluation**: the comparative experiment of seven clustering algorithms is carried out with image super-pixel (RGB value) data points as the object.

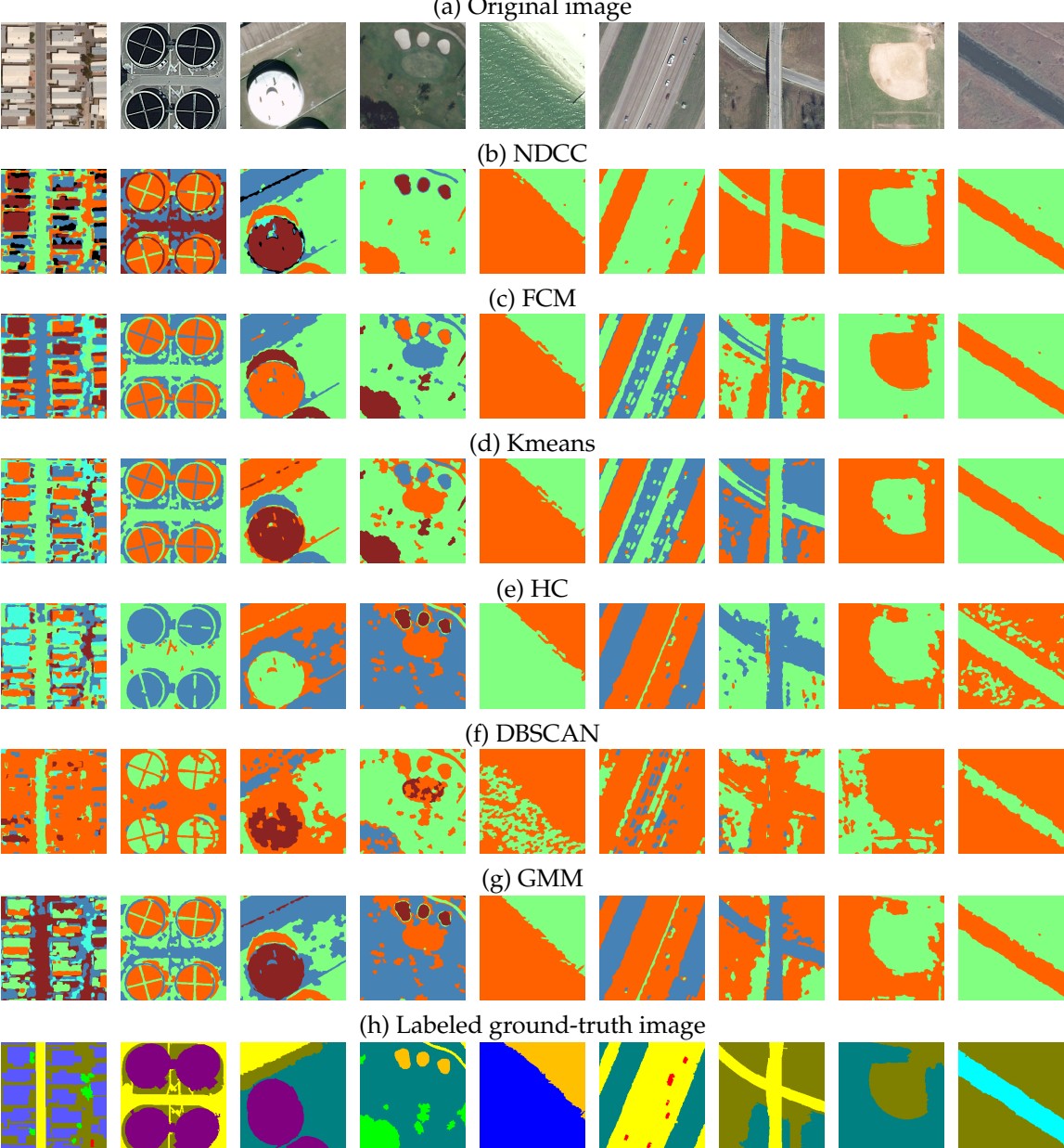

**Figure 6.** Clustering segmentation effect of six algorithms on the'UCMerced-LandUse' remote sensing dataset. Our algorithm accurately separates different features.

The 'UCMerced-LandUse' remote sensing dataset is used for verifying the algorithm clustering effect. It is a 21-class land-use-image dataset that is meant for research purposes. There are 100 images for each of the following classes. Each image measures 256 × 256 pixels. The images are manually extracted from larger images in the USGS National Map Urban Area Imagery collection for various urban areas around the country. The pixel resolution of this public domain imagery is one foot. This experiment compared the clustering effects of various algorithms cited in this paper on the dataset and verified the different clustering effects with indexes. 80 images of 21 class of ground objects are randomly selected for cluster comparison and repeated for 30 times. Table 3 shows the clustering effect pairs. Clustering segmentation effect of six algorithms on the 'UCMerced-LandUse' remote sensing dataset are shown in Figure 6.

**Table 3.** Exponential performance of various methods on 'UCMerced-LandUse' dataset. The bold data are the maximum values. All program runs 30 times. Statistically significant maximum values in the table are indicated with '*'. Additionally, the mean deviation table of clustering index is shown in the following table. The table shows that NDCC achieved good results on the dataset with labels using unsupervised methods.

| Index | NDCC | FCM | Kmeans | HC | DBSCAN | GMM |
|-------|------|-----|--------|-----|--------|-----|
| *Acc* | **0.901 ± 0.075** * | 0.795 ± 0.125 | 0.810 ± 0.010 | 0.735 ± 0.208 | 0.641 ± 0.312 | 0.791 ± 0.143 |
| *NMI* | **0.923 ± 0.031** * | 0.852 ± 0.172 | 0.825 ± 0.127 | 0.531 ± 0.213 | 0.712 ± 0.079 | 0.813 ± 0.142 |
| *ARI* | **0.867 ± 0.124** * | 0.724 ± 0.201 | 0.781 ± 0.195 | 0.812 ± 0.143 | 0.532 ± 0.183 | 0.694 ± 0.215 |
| *RI* | **0.927 ± 0.052** * | 0.885 ± 0.105 | 0.865 ± 0.083 | 0.748 ± 0.134 | 0.751 ± 0.182 | 0.742 ± 0.204 |
| *JI* | **0.846 ± 0.102** * | 0.805 ± 0.172 | 0.735 ± 0.134 | 0.624 ± 0.117 | 0.593 ± 0.157 | 0.782 ± 0.141 |

The '2015 high-resolution remote sensing image of a city in southern China' dataset of the CCF Big Data competition is used as the dataset for verifying the algorithm clustering effect. It included 14,999 original geological remote sensing images and ground-truth images, with a size of 256 × 256 pixels. Because all images of the data set are not divided, in order to better verify the clustering discrimination of the five algorithms, we randomly selected 14,000 images and divided them into 20 groups with 700 sample images each. Executing 30 times clustering in order to generate 30 groups of comparative data of different algorithms. The clustering effect pairs are shown in Table 4. The average running times are shown in Table 5.

**Table 4.** Exponential performance of various methods on '2015 high-resolution remote sensing image of a city in southern China' dataset. The bold data are the maximum values. All program runs 30 times. Statistically significant maximum values in the table are indicated with '*'. Additionally, the mean deviation table of clustering index is shown in the following table. The table shows that NDCC achieved good results on the dataset with labels while using unsupervised methods.

| Index | NDCC | FCM | Kmeans | HC | DBSCAN | GMM |
|-------|------|-----|--------|-----|--------|-----|
| *Acc* | **0.863 ± 0.052** * | 0.826 ± 0.116 | 0.825 ± 0.082 | 0.715 ± 0.092 | 0.697 ± 0.102 | 0.833 ± 0.112 |
| *NMI* | **0.815 ± 0.127** * | 0.782 ± 0.213 | 0.771 ± 0.281 | 0.655 ± 0.172 | 0.549 ± 0.098 | 0.719 ± 0.113 |
| *ARI* | **0.781 ± 0.130** * | 0.624 ± 0.142 | 0.592 ± 0.284 | 0.461 ± 0.297 | 0.464 ± 0.214 | 0.646 ± 0.194 |
| *RI* | **0.834 ± 0.128** * | 0.851 ± 0.164 | 0.812 ± 0.134 | 0.715 ± 0.161 | 0.801 ± 0.153 | 0.576 ± 0.215 |
| *JI* | **0.855 ± 0.084** * | 0.771 ± 0.213 | 0.759 ± 0.105 | 0.516 ± 0.130 | 0.499 ± 0.120 | 0.749 ± 0.171 |

**Table 5.** Comparison of runtimes for various algorithms.

| | NDCC | DBSCAN | FCM | K-Means | GMM | HC |
|---|------|--------|-----|---------|-----|-----|
| Mean time (s) | 0.6243 | 0.4937 | 0.5434 | 0.6423 | 0.756 | 0.8321 |
| Environment | CPU: AMD Ryzen2700X, eight-core processor, $f$ 3.70 GHz RAM: 16.0 GB Operating system: Windows 64-bit GPU: NVIDIA GTX1070, 8 GB GDDR5 | | | | | |

The runtime of the PDFC algorithm is not significantly different from that of other fast clustering algorithms.

### 3.4. Discussion of Experimental Results

As can be seen from Table 2, in terms of positive indicators, *NMI*, *ARI*, *RI*, and *HI* are the four indicators, while NDCC is only in Com. The data set was slightly lower than DBSCAN, and it achieved the maximum value of the other six data sets, which was the best result. The main reason was that when we used DBSCAN to verify the data, we selected the optimal result after several rounds of manual adjustment.

Besides, the Compound data set shape made it suitable for density clustering. In other data sets, despite multiple rounds of manual tuning, other methods are still unable to surpass the clustering effect of NDCC, which is completely adaptive without manual tuning. In terms of inverse indexes, *MI* indexes all obtained minimum values, indicating that NDCC can quickly find the optimal cluster on several scatter data sets. As can be seen from Table 4, when compared with the seven algorithms, NDCC is medium in terms of running time. Its running speed is generally better than that of Kmeans, FCM, and GMM, and slightly slower than that of DBSCAN and HC. However, the overall difference is order of magnitude of $10^{-2}$ s, which basically does not affect the running speed of clustering algorithm.

By randomly extracting the images from the datasets to execute 30 times clustering with the six algorithms, the measuring the mean value and deviation are presented in Tables 3 and 4. Through the comparison of five indicators, NDCC showed better indicators on two large remote sensing image datasets than the other five algorithms, and the mean deviation are not obvious when compared with other algorithms. It can indicate that NDCC had better robustness for different images. Through this statistical test, we can fairly verify the clustering effect comparison between NDCC and other algorithms on remote sensing images.

## 4. Conclusions

In this paper, we proposed the NDCC algorithm, which is a clustering method that is based on the local density of data points. As the experimental results in Table 3 and Figure 4 show, NDCC achieved the best clustering results on seven datasets, such as Flame and Aggregation. Our algorithm obtained its results without any supervision. In contrast, the other algorithms obtained relatively good results while using manually adjusted parameters. Moreover, the algorithm is further evaluated clustering effect on the 'UCMerced-LandUse' remote sensing dataset and '2015 high-resolution remote sensing image of a city in southern China' dataset remote sensing. The *Acc* and *NMI*, *ARI*, *MI*, *HI*, *RI*, and *JI* coefficients obtained showed that the clustering effect of the proposed method is better than that of five other existing algorithms. On the other hand, as the time complexity of the algorithm is at a general level, the calculation time is relatively long when processing extremely large datasets (over 100,000 data points). For each data point, we focus on the neighborhood points; hence, it is not necessary to calculate the distance between the data points that differ overly much. We can expect NDCC to perform well in natural language processing and text clustering [39,40].

In future work, we plan to optimize the structure of the algorithm according to the neighborhood characteristics of the data points, omit the calculation of the distance between data points with large differences, and reduce the time complexity of the algorithm.

**Author Contributions:** Z.W. and J.J.; methodology, software, validation, writing–original draft preparation, formal analysis, Z.L.; investigation, resources, W.L.; writing–review and editing, X.L.; project administration, funding acquisition. All authors have read and agreed to the published version of the manuscript.

**Funding:** This research was funded by Sichuan Provincial Department of the Science and Technology Program of China-Sichuan Innovation Talent Platform Support Plan (2020JDR0330) and The APC was funded by this project (2020JDR0330).

**Conflicts of Interest:** The authors declare no conflict of interest.

## Appendix A

**Table A1.** Summary of notations.

| | |
|---|---|
| $S$ | Ordered distance matrix. |
| $A$ | Ordered serial number matrix. |
| $A'$ | The core point's ordered serial number matrix. |
| $\overline{D_m}$ | The mean of the distance to the $m$-th point in the overall ordered distance matrix $S$. |
| $D_{ij}$ | The distance between $x_i$ and $x_j$. |
| $x_i$ | The $i$-th point. |
| $x_j$ | The $j$-th point. |
| $m$ | The threshold used to distinguish core, edge and noise points. |
| $E_i$ | Edge point. |
| $C_i$ | Core point. |
| $B_i$ | Noise point. |
| $N_i(x_i)$ | The number of neighborhood points of the $i$-th point. |
| $J$ | The objective function. |
| $G$ | The number of clusters. |
| $k$ | The number of the neighborhood core points which used to objective function evaluation. |
| $P(x)$ | The probability distribution function of $X$. |
| $P(y)$ | The probability distribution function of $Y$. |
| $P(x,y)$ | The Joint probability distribution of $X$ and $Y$. |
| $\delta$ | The indicator function. |
| $Y$ | Random variable named $Y$. |
| $X_i$ | The $i$-th Random variable named $X$. |
| $Y_j$ | The $j$-th Random variable named $Y$. |
| $N$ | The number of random variables. |
| $MI(X,Y)$ | The relative entropy of the joint distribution $P(x,y)$. |
| $Acc$ | The Clustering Accuracy. |
| $NMI$ | The Normalized Mutual Information. |
| $ARI$ | The Adjusted Rand Index. |
| $RI$ | The Rand Index. |
| $MI$ | The Mirkin Index. |
| $HI$ | The Hubert Index. |
| $JI$ | The Jacarrd Index. |

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
