# Peer review of "Unsupervised Clustering of Neighborhood Associations and Image Segmentation Applications"

_algorithms, doi:10.3390/a13120309_

Round 1

Reviewer 1 Report

This manuscript presents an clustering of neighborhood
associations approach with application to image segmentation.
While the article is scientific sound, some issues should be addressed
before it can be accepted for publication:

- The main advantages and disadvantages of the proposed methodology
should be presented.

- Samples datasets are shown. If there is any real dataset that can lead to the distributions that are shown in the figure. This is why they seem to be very larticular distributions.

- There is no need to define traditional cluster quality indexes.

- Summary of notations could go to the appendix.

- In Table 2 it is not clear if there should be any error
analysis included. How could one say that the differences in
performances are statistically significant?

- The authors should mention related literature
on parameter analysis, including those that studies
the influence of parameter optimization, e.g.
doi: 10.1371/journal.pone.0210236

Conclusions:

- It would be interesting, perhaps, the the proposed approach
could be used in other areas, including in language natural
processing approaches where density clusters appears:
See and mention e.g.: doi: 10.1209/0295-5075/98/58001
and "Proceedings of the ACM SIGIR: SWSM 63 (2011)".

Reviewer 2 Report

The authors have provided a deep experimental work that covers different aspects of the introduced method NDCC. Besides that, there are some minor remarks.

Clustering methods should be somehow described. I am not sure if all readers are experts in this domain.

Punctuation marks after the equations should be kept right as in the ordinary text.

P(i)(j)? -> P(x_i, x_j)?

Fig.5(c): some additional captions explaining dependencies on the graph are required.

 Fig.6(b): NDCC has not detected a cluster in the fourth figure (unlike the rest of the clustering methods). Obviously, the explanation is missing here (why it has happened).

The discussion on the obtained results is missing in the paper. The results must be discussed in detail and concluded. The authors have provided a vast range of experimental results. It is also interesting to read the authors' conclusions on some specific results (for example, I have mentioned one case above).

The first sentence in Section 2.4. does not have any sense. Please correct that.

Reviewer 3 Report

This paper proposes a neighborhood density correlation clustering algorithm that quickly discovers arbitrary shaped clusters.

The work is deeply inspired by density-based clustering. The paper is well written and presents good technical quality and a sufficient amount of qualitative and quantitative results using different performance metrics.

I have the following concerns, which can be addressed in a revised version:

- The proposed clustering method achieved the best clustering results on seven toy datasets, as well as competitive results in remote sensing datasets.
I think that the remote sensing results are the most interesting. However, it is not clear how images are processed. Is the image flattened to be represented as a vector? Some technical details should be reported.

- The related work section is quite narrow. I recommend adding recent works on distributed algorithms for density-based clustering (DOI: 10.1186/s40537-019-0207-2, 10.1016/j.engappai.2019.01.006) and remote sensing data analysis (DOI: 10.3390/rs12203276, 10.3390/s20143906)

- The results presented focus on the accuracy and correctness of the clustering, but do not report execution times of the different algorithms. Would it be possible to have summary statistics about execution times? If not, is it possible to report the upper limit computational complexity of the algorithm, in comparison with the competitor methods?

- Row 281 - 258: I think the steps should be emphasized as an itemized list or in bold.

- Can the algorithm implementation be publicly shared? This would increase the interest in the method and stimulate future research.

- I suggest replacing occurrences of "aspheric" in "spherical" since it reflects the standard terminology in data mining when referring to clustering algorithms

Round 2

Reviewer 1 Report

All issues have been addressed. My recommendation is to accept this manuscript in the present form (with minor language edits). 

Reviewer 3 Report

The authors addressed all of my concerns in a compelling manner. 

The new version of the paper suits the quality standard of the journal.

For this reason, I recommend the paper to be accepted for publication.